# A Hierarchical Clustering Method to Repair Gaps in Point Clouds of Powerline Corridor for Powerline Extraction

**Yongzhao Fan [1], Rong Zou [1,\*], Xiaoyun Fan [2], Rendong Dong [1] and Mengyou Xie [1]**

[1] Hubei Subsurface Multi–Scale Imaging Key Laboratory, Institute of Geophysics & Geomatics, China University of Geosciences, Wuhan 430074, China; YzFan@cug.edu.cn (Y.F.); 20141002074@cug.edu.cn (R.D.); xiemengyou@cug.edu.cn (M.X.)

[2] GNSS Research Center, Wuhan University, Wuhan 430079, China; 2017286180031@whu.edu.cn

\* Correspondence: zourong@cug.edu.cn

**Abstract:** Powerline detection is becoming a significant issue for powerline monitoring and maintenance, which further ensures transmission security. As an efficient method, laser scanning has attracted considerable attention in powerline detection for its high precision and robustness during the night period. However, due to occlusion and varying point density, gaps will appear in scans and greatly influence powerline detection by over–clustering, insufficient extraction, or misclassification in existing methods. Moreover, this situation will be worse in terrestrial laser scanning (TLS), because TLS suffers more from gaps due to its unique ground–based scanning mode compared to other laser scanning systems. Thereby, this paper explores a robust method to repair gaps for extracting powerlines from TLS data. Firstly, a hierarchical clustering method is used to extract the powerlines. During the clustering, gaps are repaired based on neighborhood relations of powerline candidates, and repaired gaps can create continuous neighborhood relations that ensure the execution of the clustering method in return. Test results show that the hierarchical clustering method is robust in powerline extraction with repaired gaps. Secondly, reconstruction is performed for further detection. Pylon–powerline connections are found by the slope change method, and powerlines with multi–span are successfully fitted using these connections. Experiment shows that it is feasible to find connections for multi–span reconstruction.

**Keywords:** TLS; powerline detection; gaps repair; hierarchical clustering; connection finding





## 1. Introduction

Powerlines are a vital component of the national infrastructure. However, material and structural degradation under the cyclical loading and natural erosion will result in many potential safety hazards [1]. To address such safety concerns, workmen have to inspect and maintain powerlines in harsh environments periodically; sometimes, they have to climb the high pylons for detailed inspection. This traditional field monitoring is not only dangerous but also wasteful of manpower and material resources [2]. Thus, it is a challenge to maintain powerlines efficiently, and fast but correct powerline detection is the basic foundation.

Researchers have tried a lot of remote sensing techniques for powerline detection, such as traditional camera sensors and laser systems. Compared to traditional camera sensors, laser scanning systems are not susceptible to lighting conditions, for light detection and ranging (LiDAR) can work without ambient light [1]. What is more, LiDAR can acquire dense 3D data in the information–rich form of a point cloud, which has advantages of high precision, fast scanning speed, and the ability to obtain abundant spatial information, etc. As a result, the laser scanning technique is becoming increasingly popular in powerline extraction [3].

Broadly speaking, the laser scanning system can be categorized into air borne and terrestrial–based measurements. ALS (Airborne LiDAR Scanning) is the main method of

air borne system, which integrates LiDAR, GNSS (Global Navigation Satellite System), and INS (Inertial Navigation System) on manned or unmanned aircraft to scan aerially. ALS is widely used in powerline detection for its large scanning range, but it has relatively low accuracy because of the long distance between the targets and the scanner [4]. The main modes of terrestrial–based measurements are MLS (Mobile Laser Scanning) and terrestrial laser scanning (TLS). Compared to ALS, MLS similarly mounts LiDAR, GNSS, and INS on a vehicle. The difference with ALS is that MLS can acquire data of higher precision because it scans close to targets. However, MLS is subject to the fact that it needs to travel on roads that sometimes do not exist underneath the powerlines. TLS is a high–precision environmental sensing and measurement method, which mounts the LiDAR system on a tripod to obtain data statically, but it suffers severely from two aspects: (1) decreasing point density due to increasing distance from the scanner, and (2) occlusion caused by the presence of other objects. Considering the advantages and disadvantages of these methods, related work has verified the feasibility of extracting powerlines from laser scanning data.

Point cloud powerline extraction can be categorized into three phases: preprocessing, powerline detection, and powerline reconstruction.

Preprocessing is essential for accelerating powerline extraction by simplifying point cloud data. Jung et al. [5] and Ye et al. [6] streamlined redundant point cloud data by voxel filter; they divided the point clouds into 3D voxels and replaced points by a centroid within each voxel, which could down–sample the point cloud data without losing accuracy in powerline extraction. According to the ground truth, powerlines are high above the ground, but ground points that will hinder the extraction take a great part of the point cloud data. In order to separate ground and near–ground objects from powerlines, Yang et al. [7], Shen et al. [8], and Zhu et al. [9] structured a point cloud into subspaces and used the statistical information of points within each subspace to filter out ground points in undulating terrain. Jung et al. [5] proposed a voxel–based morphological filter to isolate the objects lower than the powerline, and they defined a certain height range to extract powerlines preliminarily. Subsequently, da silva et al. [10], Guan et al. [11], and Wang et al. [12,13] used point cloud data to obtain DEM (Digital Elevation Model), DSM (Digital Surface Model), or DTM (Digital Terrain Model), combined with height difference to remove ground and near–ground points, and related studies had proved the feasibility of these morphological methods.

In powerline detection phase, powerlines are extracted and divided into individual lines. Feature extraction is widely used to calculate the various local geometric features. Jung et al. [5] and Cheng et al. [14] implemented Principle Component Analysis (PCA) to compute three eigenvalues of small clusters that derived from adjacent points, which give a clue about the presence of linear elements (powerlines), surfaces such as building facades, and volumetric objects, and they detected powerlines based on these eigenvalues. Xu et al. [15] grouped the points in labeled clusters through maximum posterior estimate; then, they extracted powerlines according to the main direction and the distance between labeled clusters. Furthermore, machine learning is another method for powerline extraction by feature calculation. Yang et al. [3] and Jwa et al. [16] trained feature models with semantic information that extracted from neighborhood points, and they isolated powerlines from other objects with the Random Forest (RF) method. Wang et al. [13] firstly extracted multi–scale features with geometric and spatial information such as line count, cylinder, k nearest neighbors, and Sphere; then, they trained the powerline model with these features by Support Vector Machines (SVM), and finally, they extracted powerlines with the Kernel Function. Guo et al. [17] used geometry and echo information of point cloud data to generate various features and estimated parameters of the learning model. They obtained powerline points with JointBoost classifier and optimized them under contextual constraints. Alternatively, transforming point cloud data into other forms such as 2D pixels or 3D voxels is also a useful way to extract information of powerlines. Wang et al. [12], Yang et al. [7], Liu et al. [18], Grigillo et al. [19], Nasseri et al. [20], and Tilawat et al. [21] used Hough Transform (HF) or Random Sample Consensus (RANSAC) to extract powerlines

from an off–ground point cloud in a 2D image, but the extraction result was sensitive to parameters that were determined by the user. Transformed data can also be applied in the statistics–based method; Zhang et al. [22], Guan et al. [11], Shen et al. [8], Zhu et al. [9], and Liu et al. [23] separated powerlines from other objects by counting points within a 2D pixel or 3D voxel according to the density difference among powerlines, pylons, trees, and other objects. However, these statistics–based methods may have a poor performance when points are unevenly distributed. The method based on prior conditions was also carried out in powerline extraction. According to the ground truth that powerlines are suspended high on the pylon, Awrangjeb et al. [24] and McCulloch et al. [25] used the prior conditions such as pylon position to identify the powerline corridor and extract powerlines, but these approaches depend on the availability of supplemental data. In addition, region growth is another efficient extraction method to extract line features; Zou et al. [1] detected a track with a modified region growth method, and they predefined the starting points and step length to derive drift vectors as the growth direction and extracted railway accurately. Liang et al. [26] used region growth to extract a powerline based on a relationship of adjacent points in the same powerline. However, neither of the region growth methods can extract a discontinuous line for its strong dependence on neighborhood relations.

In the powerline reconstruction phase, the detected powerlines are fitted with line models for powerline identification. Considering the sagging posture, powerlines are often modeled with a second–order polynomial equation in 3D [5]. Cheng et al. [14], Awrangjeb et al. [27], Yadav and Chousalkar [4], and Lai et al. [28] used second–order polynomials to fit powerlines directly in 3D space. Alternatively, fitting can be performed based on the ground truth that the powerline appears as a straight line in the horizontal plane and a second–order polynomial line in the vertical plane. Thus, Guan et al. [11], Ortega et al. [29], and Yin et al. [30] carried out a stepwise fitting method to reconstruct the powerline in the horizontal plane and vertical plane, respectively.

Recently, studies on powerline extraction from point cloud data are relatively mature. However, due to decreasing point density and occlusion causing by the other objects in the line of sight, gaps will appear during the scanning process, resulting in uneven points distribution and discontinuous lines, which further greatly affect the performance of powerline extraction by over–clustering, insufficient extraction, and misclassification. What is worse, previous studies had few efforts in extracting powerlines by repairing gaps. To address such concern, this paper explores a hierarchical method that takes full use of neighborhood relations to repair gaps in a point cloud of powerline corridors for powerline extraction as well as a multi–span line fitting method based on pylon–powerline connection for further detection. A TLS point cloud is used as the provisionally appropriate data to validate the gap repair method, because TLS produces more occlusion than other laser scanning systems. The main experiment in this paper can be divided into following steps: First, off–ground point cloud data are cut into segments with a predefined step length along the powerline, and within each segment, points are grouped into various clusters by Euclidean Clustering (EC). Then, the centroids of these clusters are calculated, and centroids of gaps are estimated by centroids of their adjacent powerline clusters. With these estimations, powerline clusters that have the nearest centroid are clustered hierarchically into individual lines. Finally, each individual line with multi–span is fitted according to pylon–powerline connections found by slope change method. As a result, powerlines with multi–span will be directly extracted and fitted as an individual line. The main contributions of this research are as follows:

1. Explore a new method to repair gaps for powerline extraction, which solves the problem of over lustering and insufficient extraction caused by gaps in the existing method. The method has been tested in various gap situations, and experiments show that the method is of high robustness.
2. Propose a method of searching pylon–powerline connections based on the slope change and reconstruct the powerline with multi–span.

The structure of the article is as follows: Section 2 introduces the experimental data, Section 3 presents the processing flow and main algorithms, Section 4 analyzes and discusses the powerline extraction and reconstruction results, and Section 5 is the summary.

## 2. Data Description

The experimental TLS point cloud data were measured near the Land and Resources College in Hannan District of Wuhan, Hubei Province, China. Hannan District is located in the southwest of Wuhan (30.30879°N, 114.08462°E), which is an urbanized area far from the downtown with gentle terrain, and there are sparse shrubs and low buildings distributing in this area. Figure 1 displayed the location of the test field.

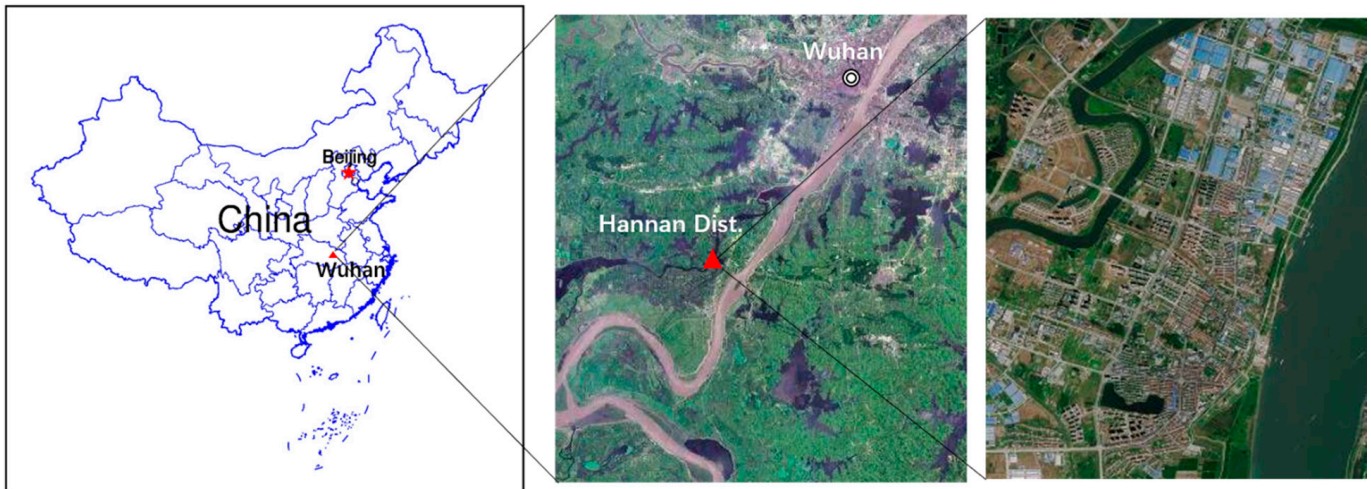

**Figure 1.** Location of the test field.

In this test field, powerlines with a transmission voltage of 35 Kv are distributed on both sides of the road. The point cloud data were obtained by the laser scanner FARO Focus 3DS 120 in 2020 and were completed with SCENE, which is a software packaged with the FARO scanner. The instrument specifications of the scanner are list in Table 1. As shown in Figure 2, the obtained TLS point cloud data have lots of gaps and contain powerlines with two spans, which is suitable for the gap repair method and multi–span reconstruction.

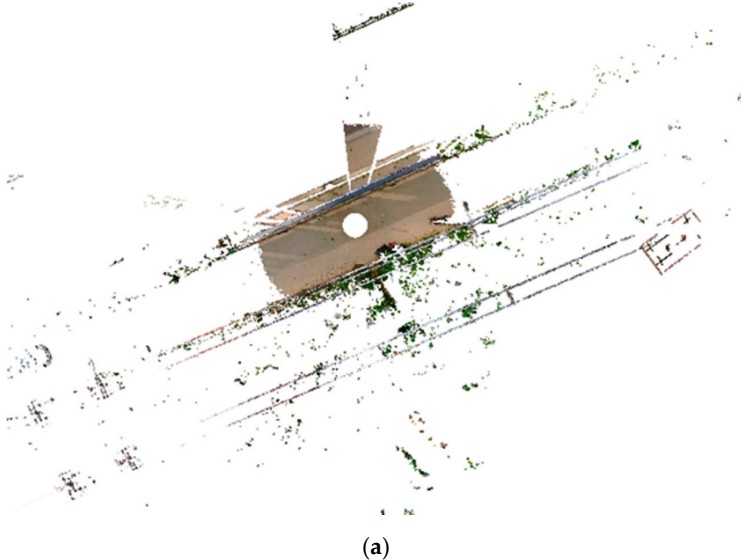

(**a**)

**Figure 2.** *Cont*.

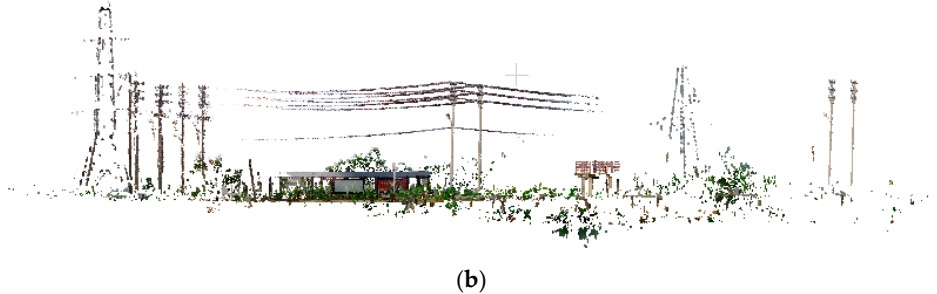

**(b)**

**Figure 2.** The original data with RGB color. (**a**) Top view of the original data; (**b**) Front view of original data.

**Table 1.** Parameters of FARO light detection and ranging (LiDAR).

| Parameters Type | Values |
| --- | --- |
| Scan radius | 120 m |
| Horizontal scan range | 300° |
| Vertical scan range | 360° |
| Ranging error | ±2 mm |
| Scan speed | 976,000 p/s |
| Color options | 70 million built–in pixels |

## 3. Methodology

As a LiDAR scanning system, TLS has the advantages that can obtain accurate three–dimensional spatial information, and it is not affected by illumination conditions. Additionally, TLS data can be used as provisionally appropriate data to verify the gap repair method because it suffers severely from the occlusion, which produces lots of gaps in scans. Figure 3 visually illustrates the workflow of the proposed approach to repair gaps and extract the powerline from the TLS data. First, the raw data are preprocessed by data cropping, ground filter, height filter, and voxel filter to refine and thin the numerous points. Second, the experimental point cloud data are divided into continuous segments, and the points of each segments are grouped into clusters by EC. Then, the centroid and slope of each cluster are calculated for further extraction. After that, a hierarchical clustering method between segments is carried out, and adjacent powerline clusters that have the nearest centroids are combined into one. In order to solve the problem of discontinuity caused by the gaps, this paper takes full use of neighborhood relations to repair gaps, and all powerlines are extracted as individual lines directly. Finally, multi–span reconstruction is performed on each line. The powerline–pylon connections that part the powerline into different spans are found by the slope change method, and then a powerline with multi–span is fitted based on these connections.

### 3.1. Data Preprocessing

In this article, point cloud data preprocessing include data cropping, ground filtering, height filtering, TLS data thinning, and data rotation. TLS point cloud data contain lots of redundant points, which will greatly slow down the powerline extraction. Thus, data cropping is necessary to reduce the point cloud data. Points that were far from the powerline corridors were removed as outliers without losing any powerline points. The cropped data contain a total of 17,159,420 points, with a horizontal span of 46 m and a vertical span of 10 m. The data include four main parts: pylons, vegetation, buildings, and powerlines with two spans. To highlight the vertical distribution of the objects in the powerline corridor, we colored a point cloud according to the elevation. As shown in Figure 4, it is obvious that most objects of the cropped data are lower than powerlines, and the test field has a relatively flat terrain.

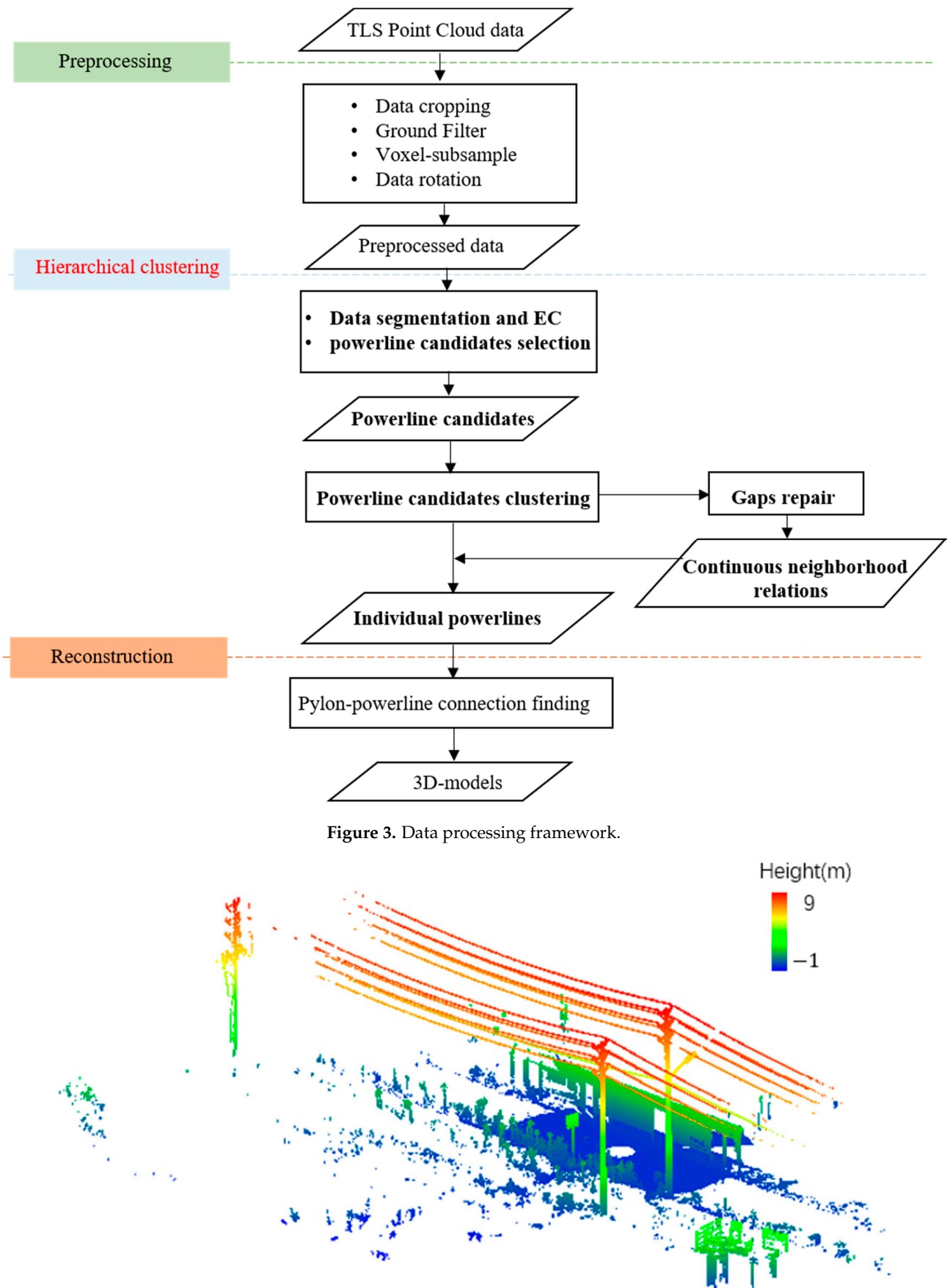

**Figure 3.** Data processing framework.

**Figure 4.** The data cropped from original terrestrial laser scanning (TLS) data.

Ground filtering and height filtering are important prerequisites to reduce unwanted objects. The point cloud data not only include powerlines but also a large number of outliers that are lower than powerlines (ground, shrubs, low buildings, etc.), which will hinder the extraction of the powerline point cloud. The cloth simulation filter is an efficient ground filtering method that utilizes the nature of cloth and modifies the physical process of cloth simulation to adapt to point cloud filtering [31]. For disconnecting the powerlines from the ground, the ground is detected and removed by cloth simulation filter in this study. However, ground–filtered data may still include some unwanted objects that are lower than the powerlines. The height filtering method that is based on the DTM (Digital Terrain Model) derived from the detected ground can further reduce the outliers and isolate the powerlines [7,8]. Thus, we also performed height filtering based on DTM. The DTM derived from the detected ground was raised to a certain height ($h_{min}$), and points that were lower than the raised ground were removed. According to the National Electrical Safety Code (NESC), the vertical clearance of the lowest–level powerline is standardized as 6.4 m in China [11]. Considering the sagging factor, $h_{min}$ is set at a value of 5 m, which is lower than the NESC standard to ensure we did not miss any powerline points. As shown in Figure 5a, points that are 5 m above the ground are further extracted as off–ground points, while others are filtered out as outliers, and the filtered data are 90% less than before with total of 134,463 points.

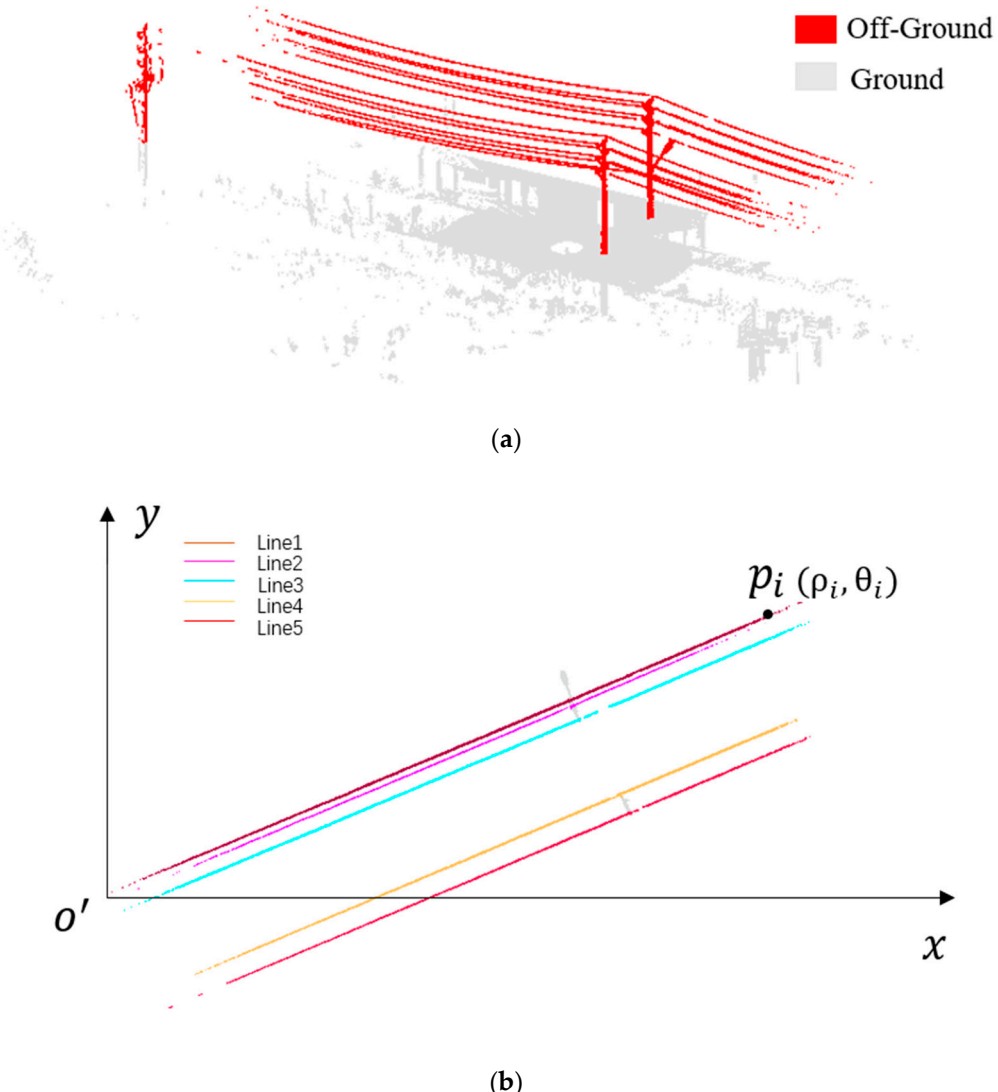

(a)

(b)

**Figure 5.** *Cont.*

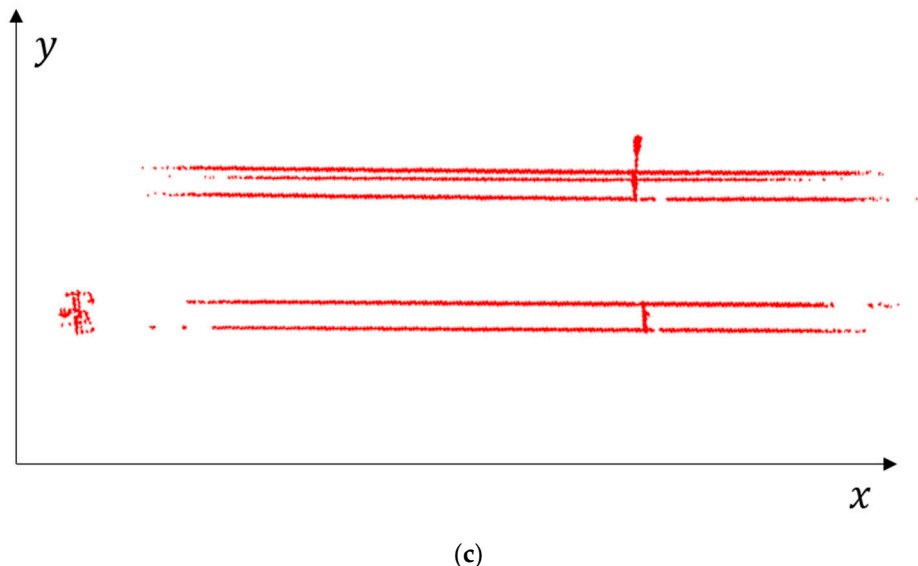

(**c**)

**Figure 5.** Data preprocessing. (**a**) Result of ground filtering and height filtering. Red points indicate the off–ground part, gray points are the ground part that has been removed. (**b**) Calculation of $\theta$. (**c**) Rotated data (top view).

The ground–filtered point cloud is of great redundancy with numerous points, which will influence the efficiency of processing. Thereby, data thinning is essential to accelerate the further processing. A voxel filter was used to streamline the ground–filtered point cloud; point cloud data were put into 3D voxels, and each voxel was 8 cm$^3$ in size; then, points within the same voxel are replaced by the centroid of them. As a result, the voxel–filtered points are reduced by 90% compared to the original data, with a total of 1,350,728 points left. However, the original structure of the test point cloud data remains without losing any accuracy.

Since the proposed method is carried out along the powerline, it is necessary to rotate the point cloud until powerlines are parallel to the $X$–axis. Powerlines are parallel to each other in XOY plane; thus, calculating the angle of one of the powerlines is enough for rotation. First, we selected line1 on the XOY plane and converted the coordinate origin to one end of line1, as shown in Figure 5b; $p_i(\rho_i, \theta_i)$ represents 2D points on the line1 ($i = 1 - n$, $n$ is number of points). When line1 is parallel to the $X$–axis, $\theta_i$ and the rotation angle $\theta$ should satisfy the following conditions:

$$\sum_{i=1}^{n} (\theta_i - \theta)^2 \geq \sum_{i=1}^{n} (\theta_i - \theta_j)^2 \tag{1}$$

where $\theta_j$ represents any value of the rotation angle.

Subsequently, we rotated the point cloud data based on the obtained $\theta$; the RMSE (Root Mean Square Error) of the rotation is 0.014 rad, and the rotated point cloud is shown in Figure 5c.

### 3.2. The Hierarchical Clustering Method

#### 3.2.1. Segmentation and Powerline Candidate Selection

In this part, preprocessed data are divided into continuous segments and powerline candidates are obtained by EC within each segment. First, preprocessed data are divided with a predefined step length $s_1$ to get continuous segments that contain the points of powerlines and pylons in Figure 6a; then, points in each segment are clustered into different groups by EC based on constraint distance $r$. Figure 6b demonstrates the EC algorithm: EC starts from a random point (colored in green), and points whose distance from the start point is less than $r$ are grouped. After that, a new start point is picked from this group (except the previous start point) and the clustering is continued with $r$ until there are no

more points that can be added. Then, we pick a new point from the rest points as the new start point and repeat the above process until all the points are clustered. Figure 6c exemplifies the results of EC with five clusters ($C_1 \sim C_5$). As the consequence of EC, the points of the powerline and pylon in each segment are grouped into series of unlabeled clusters, which are subsequently used for selecting powerline candidates.

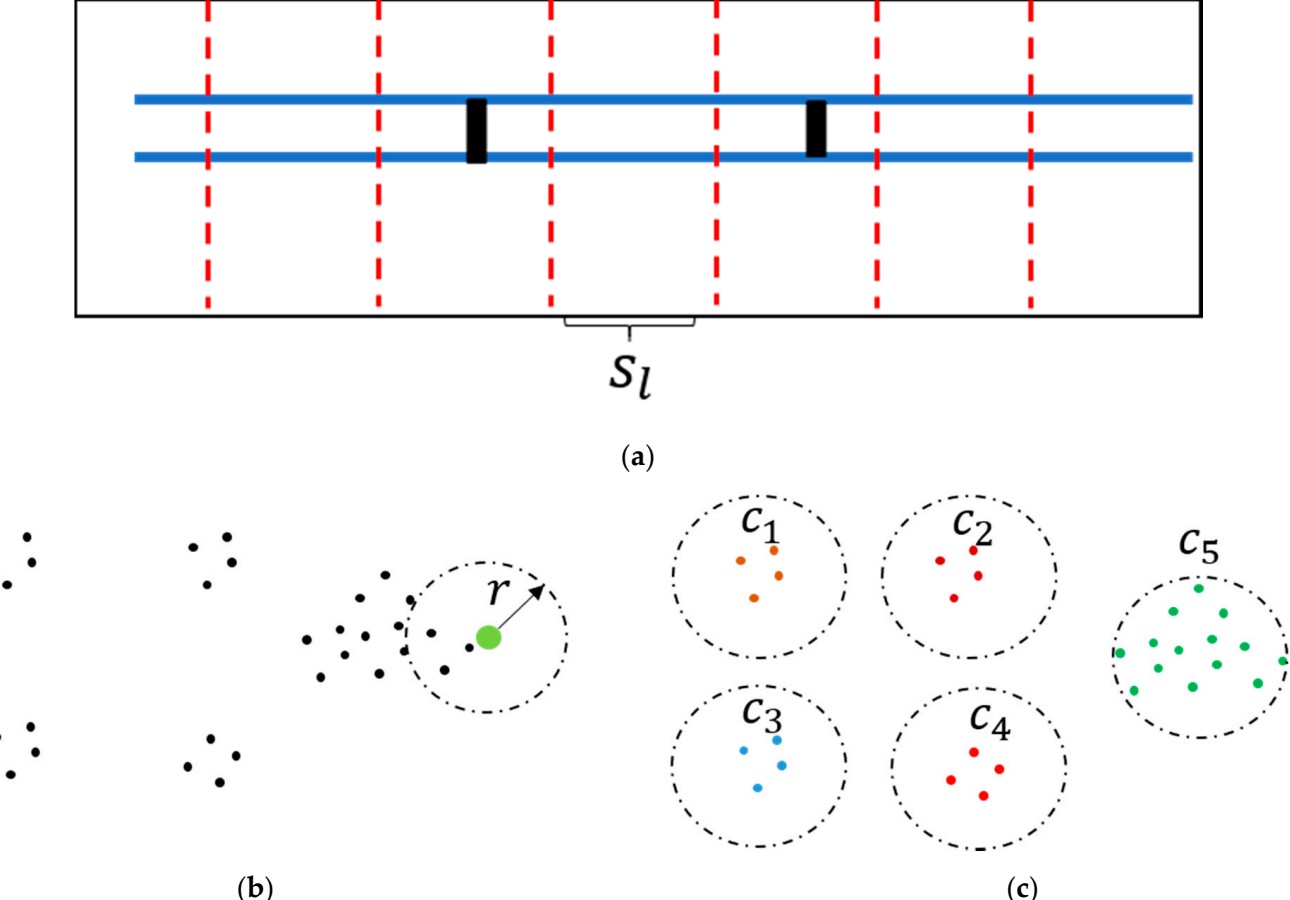

**Figure 6.** Segmentation and Euclidean Clustering (EC). (**a**) Segmentation of preprocessed data with predefined step length $s_1$. Blue lines represent powerlines and black lines represent pylons. (**b,c**) The schematic diagram of EC within a segment (cross–section).

Although various clusters are obtained by EC, clusters that represent powerlines are still unknown. Finding powerline clusters is essential for executing gaps repair and individual powerline detection. Since the powerline cluster can be regarded as an approximate cylinder and its cross–section is approximately a circle, we can distinguish the power line from other objects by the following steps:

(1) Determine the slope of all the clusters using LS (Least Square method) in plane corresponding to the direction of the powerline (in this paper, the plane is XOZ).

(2) Rotate the clusters around the Y–axis that is perpendicular to the plane of the powerline run.

(3) Distinguish powerline clusters in the cross–section. For each cluster, $sp_h$ (horizontal span) and $sp_v$ (vertical span) of the cross–section are calculated to reflect the cluster size, and clusters approximately equal to $sp_h$ and $sp_v$ (ratio of $sp_h$ and $sp_v$ larger than predefined value $sp_t$) are regarded as powerline clusters. It should be mentioned that we set $2r$ as the threshold of the horizontal and vertical span to further distinguish powerline clusters from others. Detail judgment is as follows:

$$\begin{cases} abs(sp_h - sp_v) > sp_t \\ \quad sp_v < 2r \\ \quad sp_v < 2r \end{cases} . \tag{2}$$

Consequently, the powerline clusters are labeled with "line" as powerline candidates, distinguishing from others labeled with "no–line". Figure 7b,c exemplify the powerline candidates selection in each segment.

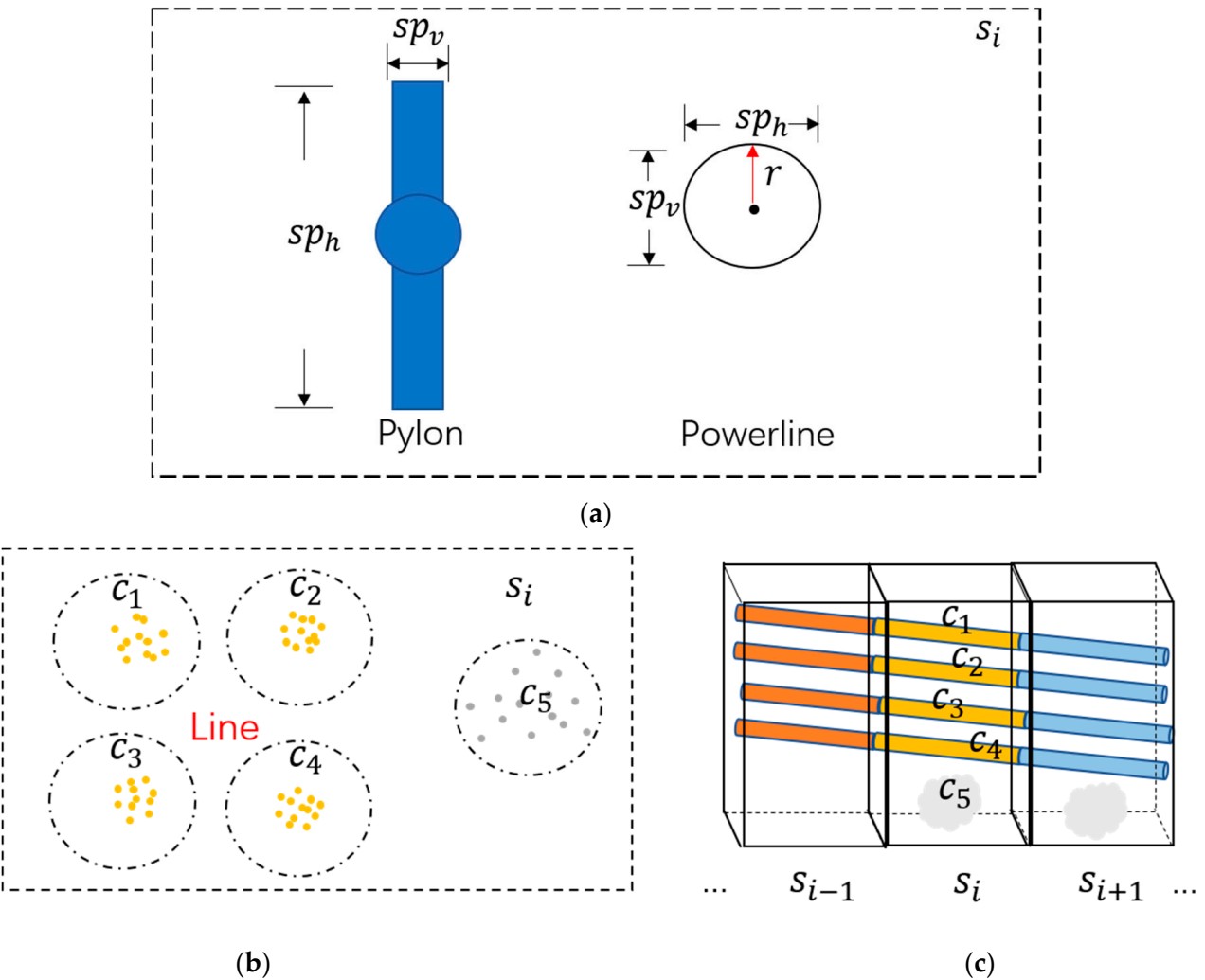

(a)

(b)                                              (c)

**Figure 7.** Powerline candidate selection. (**a**) Characteristic of rotated powerline and pylons. Black arrows denote the principal direction of different clusters. $s_l$ is the step length of segmentation, and the circle with radius $r$ is the cross-section of the powerline cluster. (**b**) Result of powerline candidate selection (cross–section), $c_1$–$c_5$ are clusters in the segment $s_i(i = 1, 2, 3 \ldots)$, $c_1$–$c_4$ are selected as powerline candidates with "line" label (**c**) Result of powerline candidate selection (front view). $s_i$ is the front view of (**b**), $s_{i-1}$ and $s_{i+1}$ are neighboring segments of $s_i$. Powerline candidates of the same segment are colored the same, clusters that are not powerline are displayed in gray.

### 3.2.2. Powerline Candidates Clustering and Gaps Repair

Based on the above steps, preprocessed data are divided into several continuous segments and powerline candidates are obtained, while the gap repair method and powerline extraction can be further implemented. In our study, we calculated centroids of powerline candidates and used distance of centroids as the clustering foundation, for powerline candidates in the same individual powerline are closely connected, which means that every

two adjacent powerline candidates in the same individual powerline have the nearest centroids compared to others.

In this work, our clustering method is to combine the powerline candidates of the same individual powerline segment by segment, and it begins with a "starting segment". First of all, the segment with the most powerline candidates is regarded as a starting segment, and its adjacent segments is called the "matching segment". Then, powerline candidates in the starting segment are labeled from 1 to $n$ ($n$ is count of these candidates), and they are matched successively along both sides of the starting segment. For either side of the starting segment, the distance between the powerline candidates of the starting segment and candidates of the matching segment are computed, candidates that have the nearest centroids are labeled the same, and candidates of the staring segment that have found their nearest candidates in the matching segment are labeled "matched". After that, the visited matching segment is set as a new starting segment to continue the clustering until all the segments are traversed. Figure 8a illustrates the clustering method visually.

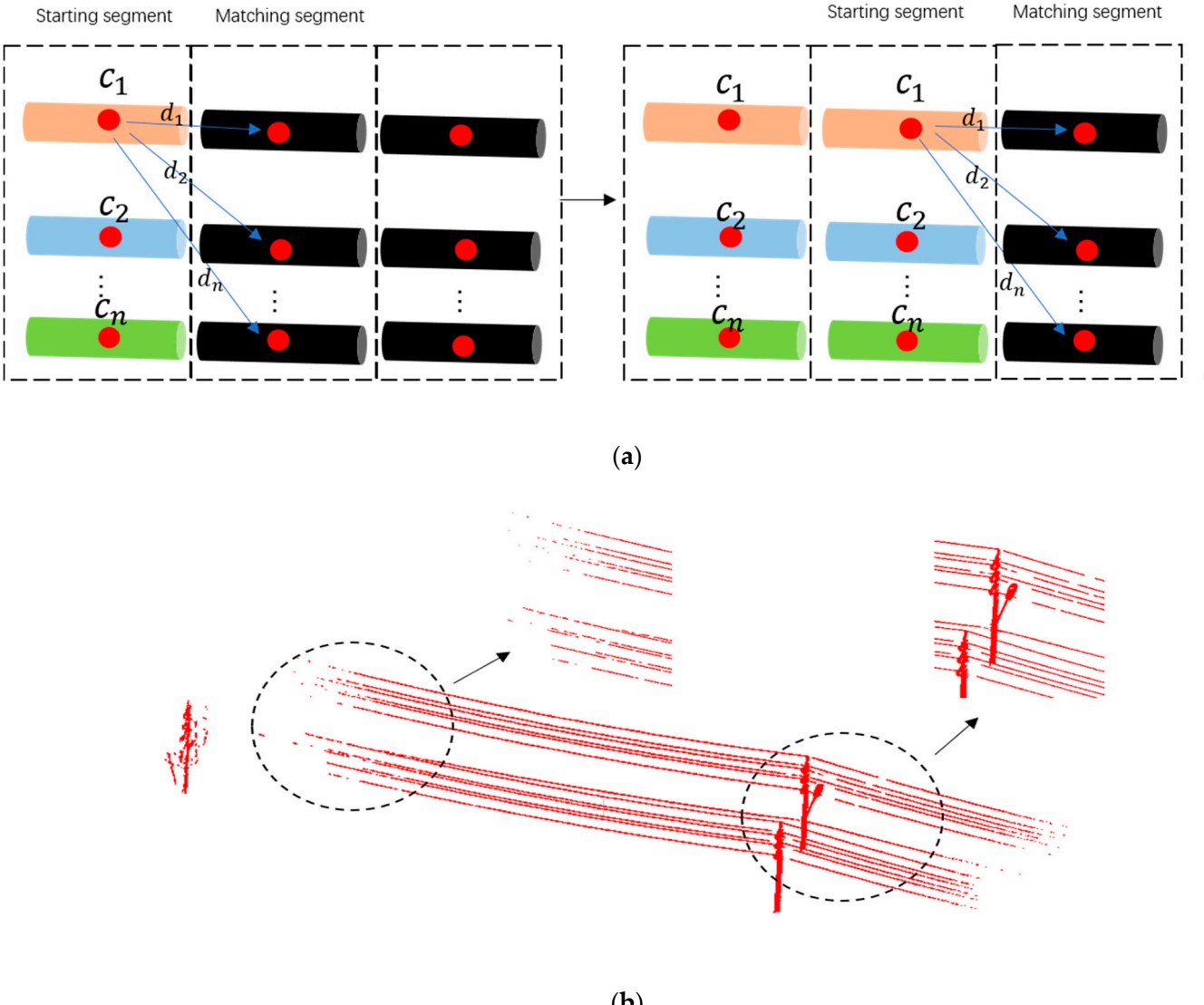

(a)

(b)

**Figure 8.** Example of candidate clustering and gaps. (**a**) Clustering between the starting segment and matching segment (take the clustering process of $c_1$ as an example. Black cylinders are powerline candidates in matching segments, $c_1-c_n$ are labeled powerline candidates in the starting segment, red points are centroids of powerline candidates, and $d_1-d_n$ are the distance between the centroids). (**b**) Gaps in the preprocessed data.

In theory, the hierarchical clustering method could extract individual powerlines directly. Unfortunately, powerline candidates are not continuous due to gaps, as shown in Figure 8b, which results in discontinuous neighborhood relations and will severely block the execution of the clustering method. Moreover, methods rely on neighborhood relations; for example, EC and the region growth method are also severely plagued by this problem. To ensure clustering, the method mentioned above is refined by the gap repair method as follows:

(1) Gaps detection. In the above hierarchical clustering, candidates without a "matched" label in the starting segment can imply that they are discontinuous in the neighborhood, indicating the existence of gaps in the matching segment. Thus, gaps can be found with these "unmatched" candidates.

(2) Centroid estimation. According to the ground truth that powerlines of the same span share the similar morphological characteristics, we can infer that centroids of powerline candidates in the same segment have similar variation tendency. Hence, centroids of gaps can be estimated to create continuous neighborhood relations. The formulas of estimation are as follows:

$$avec_{(x,y,z)} = \sum_{j=1}^{N} \frac{c^j_{i\,(x,y,z)} - c^j_{i-1(x,y,z)}}{n} \tag{3}$$

$$C^k_{i(x,y,z)} = C^k_{i-1(x,y,z)} + avec_{(x,y,z)}. \tag{4}$$

In Formula (1), the average change in centroids of matched candidates (avec) is computed as the estimated change in centroids of unmatched candidates. $c^j_{i\,(x,y,z)}$ and $c^j_{i-1(x,y,z)}$ respectively denote the matched powerline candidates of the adjacent segment and starting segment, $i$ and $i-1$ represent two adjacent segments (starting segment and matching segment), and n is the count of the matched candidates in the starting segment. In Formula (2), centroids of gaps are estimated by centroids of unmatched candidates and avec. $C^k_{i(x,y,z)}$ is the estimated centroid of the gap, $C^k_{i-1(x,y,z)}$ is the unmatched candidate in the starting segment, and k is the label of candidates that have not found their nearest candidates in the matching segment. However, it is impossible to obtain avec if there are not any powerline candidates in the matching segment. To improve the robustness, we used *avec* of the previous adjacent segments for estimating centroids of gaps. Figure 9 comprehensively demonstrates the method of gap repair.

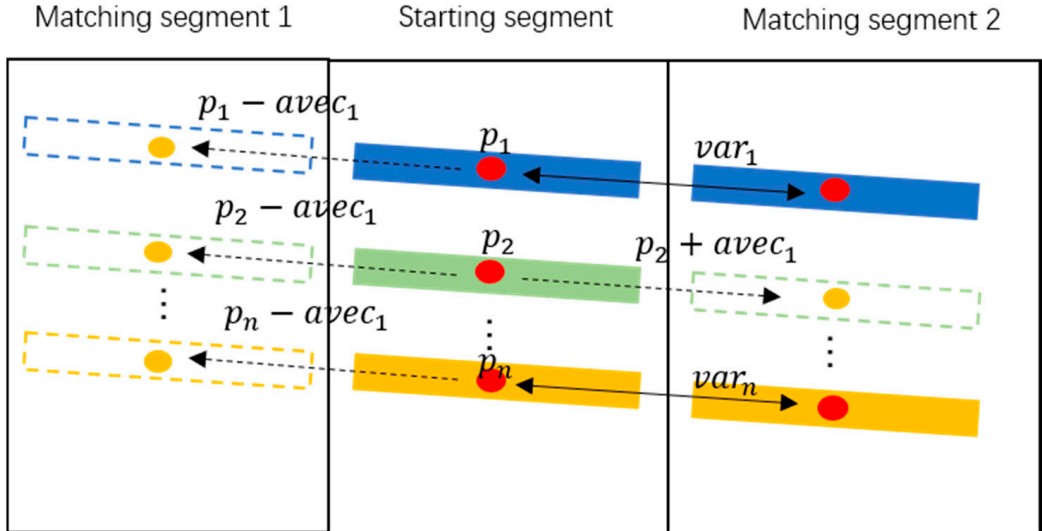

**Figure 9.** Gap repair method. Red circles are centroids of matched powerline candidates, orange circles are estimated centroids. Colored rectangles represent powerline candidates, and dotted rectangles represent gaps. $p_1 - p_n$ denote centroids of candidates in the starting segment. $var_1 - var_n$ are the changes in centroids of the matched powerline candidates between the starting segment and matching segment1, $avec_1$ is the average value of them.

In summary, gaps repair and powerline candidates clustering are carried out simultaneously. Gaps are repaired based on the clustering method; in return, repaired gaps facilitate the execution of clustering. As a result, neighborhood relations are continuous and powerline candidates are grouped into an individual line directly in the case of gaps.

### 3.3. A Powerline Connection Finding Method Based on Slope Change

The powerline has multiple forms in different spans. Therefore, it is difficult to fit a powerline with multi–span using only one model. In this article, pylon–powerline connections are found by the slope change method; then, powerlines of different spans are modeled and connected with these connections. Finally, powerlines with multi–span can be successfully fitted. Slopes of powerline candidates in the XOZ plane are computed in Section 3.2, and they are further used to find the connection. Pylon–powerline connections lie between the adjacent powerline candidates that have the largest slope change. Thus, as shown in Figure 10, we firstly used the slope change method to find the two adjacent powerline candidates that have the largest slope change (indicate the position of the pylon–powerline connection); then, a quadratic polynomial is used to fit the candidates respectively, and the pylon–powerline connection is found by computing the intersection of them. It should mention that we used the proposed method to find connection in the XOZ plane, and the y value is estimated by a straight line model in the XOY plane.

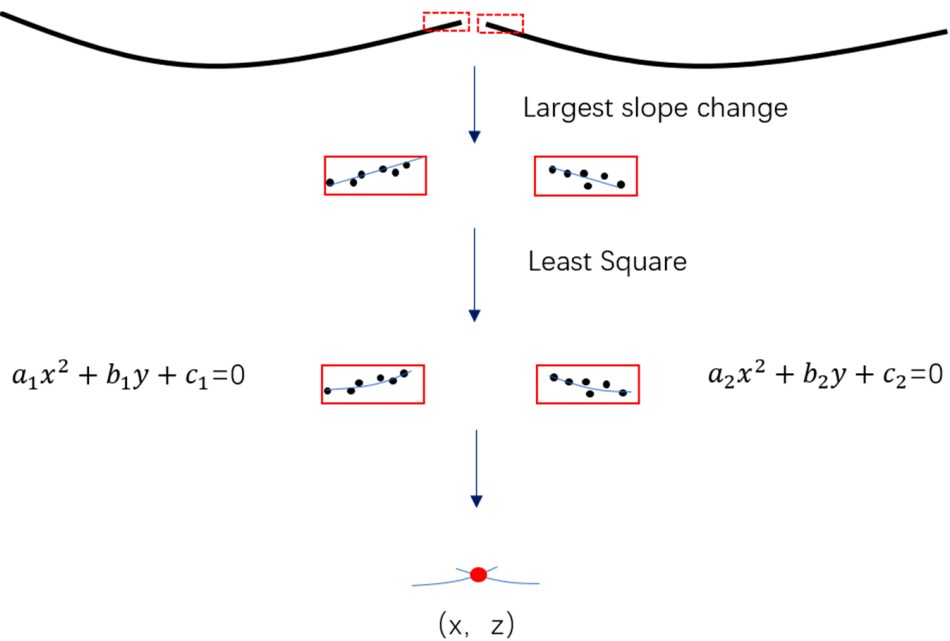

**Figure 10.** Powerline connection finding based on slope change. Red regions represents the adjacent segmentations that have the largest variation of a single powerline, and (x,z) is the pylon–powerline connection.

## 4. Results and Discussion

This section mainly shows the results of the proposed hierarchical clustering method based on gap repair and the results of multi–span powerline reconstruction. Table 2 shows strategies for powerline extraction and reconstruction, and the thresholds are discussed in Section 4.3.

### 4.1. Powerline Extraction

In order to deal with the problems caused by gaps, this article proposed a robust and automatic algorithm framework for extracting powerlines from point cloud data. As shown in Figure 11b,c, gaps are distributed disorderly in the preprocessed data. In this case, continuous neighborhood relations are created by the gap repair method, and

powerlines are extracted from preprocessed data and divided into individual lines, as shown in Figure 11a. The result shows that the hierarchical clustering method proposed in this paper can effectively solve the problem caused by gaps, and individual powerlines can be extracted from preprocessed data directly without any prior conditions.

**Table 2.** Parameters in proposed method.

| Parameter | Description | Values |
|---|---|---|
| $s_l$ | Step length | 0.5 m |
| $r_d$ | Constraint distance of EC | 0.3 m |
| $r$ | Radius of powerline candidates | 0.02 m |
| $sp_t$ | Ratio of vertical span and horizontal span | 0.7 |

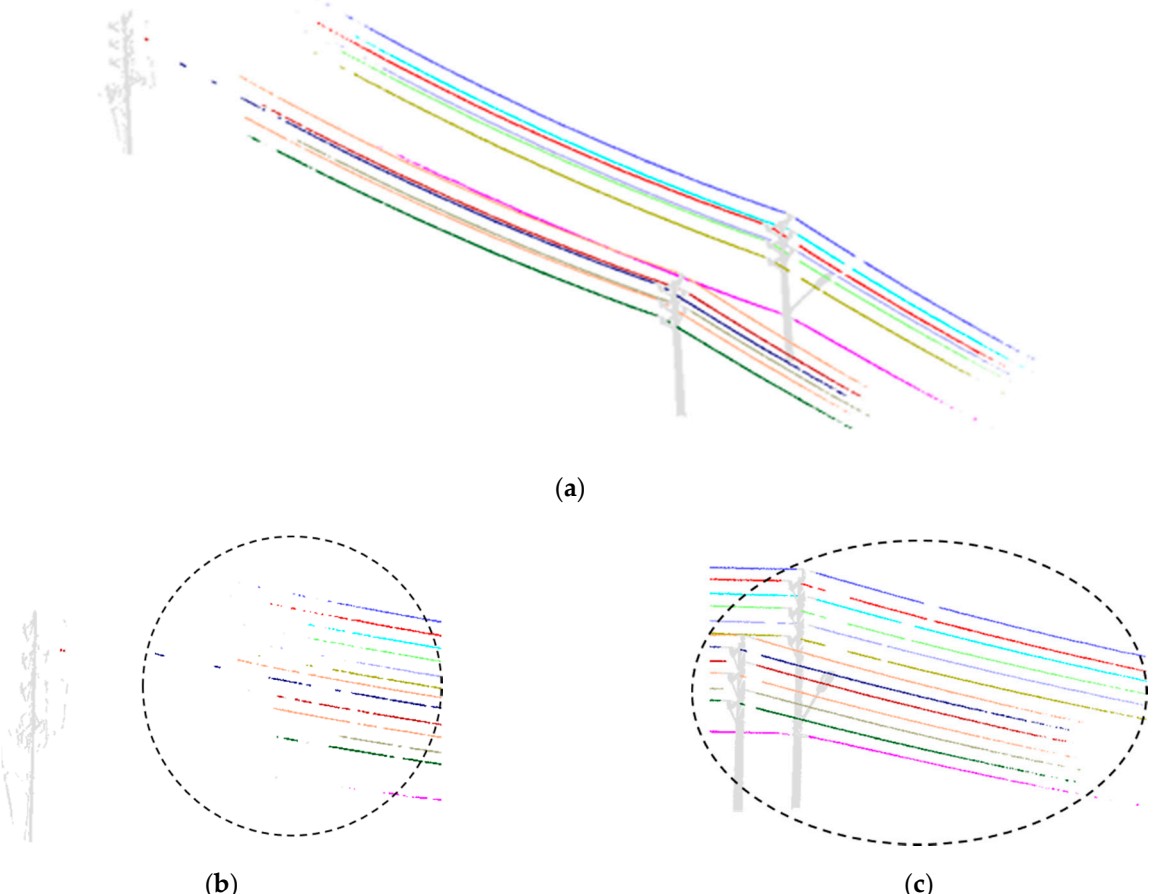

(**a**)

(**b**)　　　　　　　　　　　　　　　　　(**c**)

**Figure 11.** The results of powerline extraction. (**a**) Each single powerline is successfully extracted from preprocessed data directly and shown in different colors. (**b**,**c**) Clustering with the gaps. Powerlines are clustered correctly.

Although extracting powerlines from data containing gaps has been proved in preprocessed data, gaps not only distribute disorderly but also appear as a whole segment in a powerline point cloud. To further verify the robustness of the hierarchical clustering method, we manually removed the entire segment of the powerline to simulate the gaps and continuously enlarge the segment width to test the ability of this method in repairing gaps. As shown in Figure 12, data with segment width of $s_l$ (step length), $2s_l$, $4s_l$, $8s_l$, and $12s_l$ are used as the test data to validate the robustness of the hierarchical clustering method. Experimental results show that proposed method can extract individual powerlines within the segment width $12s_l$. When the segment width is larger than $12s_l$, different individual lines will be grouped into one due to the accumulation of errors in centroid estimation.

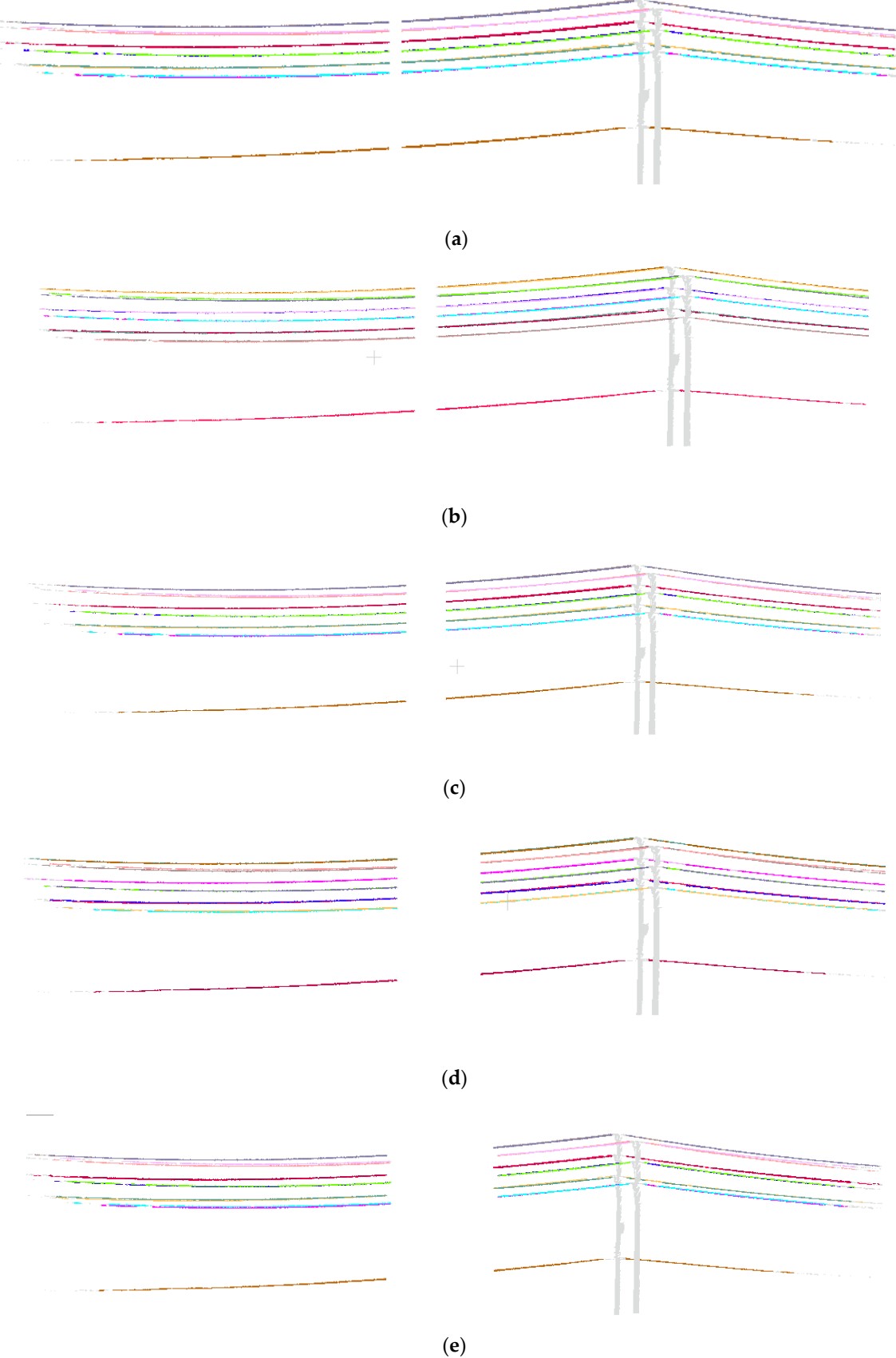

**Figure 12.** Robustness test. (**a**) Individual powerlines extraction with segment width $s_l$. (**b**) Individual powerlines extraction with segment width $2s_l$. (**c**) Individual powerlines extraction with segment width $4s_l$. (**d**) Individual powerlines extraction with segment width $8s_l$. (**e**) Individual powerlines extraction with segment width $12s_l$.

The proposed method has been proved robust under data with various gaps. However, some powerline points are treated as outliers that were distributed close to the pylons in Figure 13. We conclude that the points close to pylons are grouped with pylons during the extraction. Although these points only take an extremely small part of powerlines, in order to seek for greater perfection, we set step length empirically to get powerline points closed to pylons as much as possible and discuss it in Section 4.3.

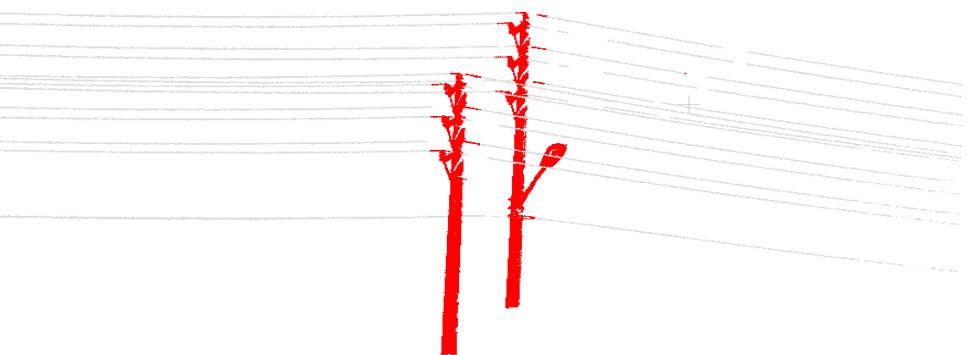

**Figure 13.** The missing powerlines points close to the pylons.

Related work has also made many efforts in powerline extraction. Region growth is an efficient method for line extraction because it can take full use of neighborhood relations; Zou et al. [1] used the modified region growth method by deriving drift vectors, and they extracted railway accurately in the case of bends and a straight line. Liang et al. [26] implemented traditional region growth for powerline extraction, and they extracted the powerline completely from point cloud data. However, for every plus, there is a minus. The region growth method, which takes full use of neighborhood relations, also relies on neighborhood relations strongly. When gaps increase in powerline point cloud data, the region growth method may have poor performance in powerline extraction and results in insufficient extraction, as shown in Figure 14a. Guan et al. [11] conducted EC combined with HF to detect transmission lines in a rural environment. Unfortunately, when the neighborhood relations are not continuous, the optimal constraint distance of EC may not exist. If the constraint distance is relatively small, EC may over–cluster powerlines into multiple parts in Figure 14b, but if we enlarge the constraint distance, it will misclassify different individual powerlines by merging them into the same clusters, as shown in Figure 14c.

Some other methods have a certain tolerance to the gap. The method proposed by Awrangjeb et al. [24] is less affected by gaps, but it relies too much on positions of pre–detected pylons, which makes the extraction less automatic. Jung et al. [5] and Cheng et al. [14] used a morphological method to extract individual powerlines. They calculate the RMSE of the fitting model of powerline clusters in the XOY plane and XOZ plane, as well as the distance between the end points of the powerline clusters to obtain individual powerlines. To a certain extent, this method is able to deal with gaps, but it contains a large amount of calculation, and the constraints are complicated. In addition, its ability to handle large gaps remains to be verified. Yang et al. [7] proposed an improved HT method combined with line model to detect powerlines, but the accuracy would be affected by large gaps and the distance of the transmission corridor.

Based on the above results and analysis, the proposed method has been proved robust in individual powerline extraction, which is also an improvement in handling large gaps compared to the existing methods.

### 4.2. Powerline Recosntruction

Then, powerline reconstruction is carried out for further detection. According to the geometry of the powerline, our experiment performs least squares fitting on two 2D planes to reconstruct the 3D powerline, the geometry of a powerline is sufficiently appropriate

modeled as a straight line on horizontal plane (5) and a quadratic polynomial line on vertical plane (6), where *a*, *b*, and *c* denote the parameters of the equation of a straight line or polynomial line, while x, y, and z denote the 3D coordinates of the powerline points:

$$ax + by + c = 0 \qquad (5)$$

$$z = ax^2 + bx + c. \qquad (6)$$

Different from previous studies, we fit the powerline with multi–span based on connections found by the slope change method. As shown in Figure 15a, connections are successfully estimated, they are located in the middle of adjacent powerline parts that have the largest slope change. Using the obtained connections, the powerlines can be fitted preferably with two spans in Figure 15b. Although powerlines in test data only have two spans, it can verify the feasibility of the pylon–powerline connection finding method. Furthermore, a powerline with multi–span can also be fitted based on the connection.

### 4.3. Parameter Setting

#### 4.3.1. Step Length Setting

The size of the step length affects the morphological characteristics of the powerline clusters in each segment. After several experiments, the step length is set to 0.5 m, which is greater than width of the pylon along the *X*–axis (0.4 m). This step length can not only separate all the pylon points from test data but it can also maintain the morphological characteristics of powerline clusters. If the step length is given a higher value, centroids of powerline clusters may not be on the line, which will result in deviation in the hierarchical clustering method. If the step length is given a lower value, pylon points may be mixed into the powerline points, which will affect the powerline candidate selection.

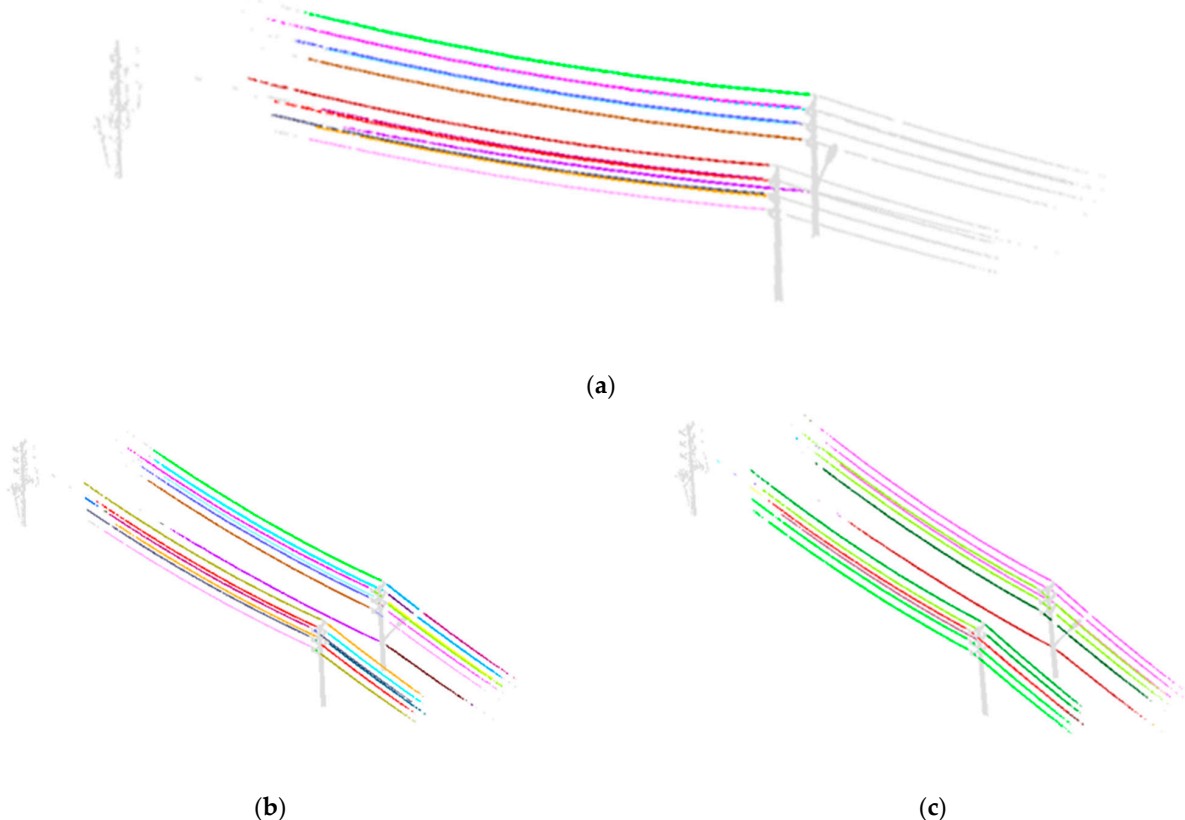

(**a**)

(**b**)                                                                 (**c**)

**Figure 14.** Using region growth and EC to extract powerlines from preprocessed data. Individual powerlines are distinguished by different colors. (**a**) Insufficient extraction of the region growth method. (**b**) Over–clustering of EC due to the small constraint distance. (**c**) Misclassification of EC due to the large constraint distance.

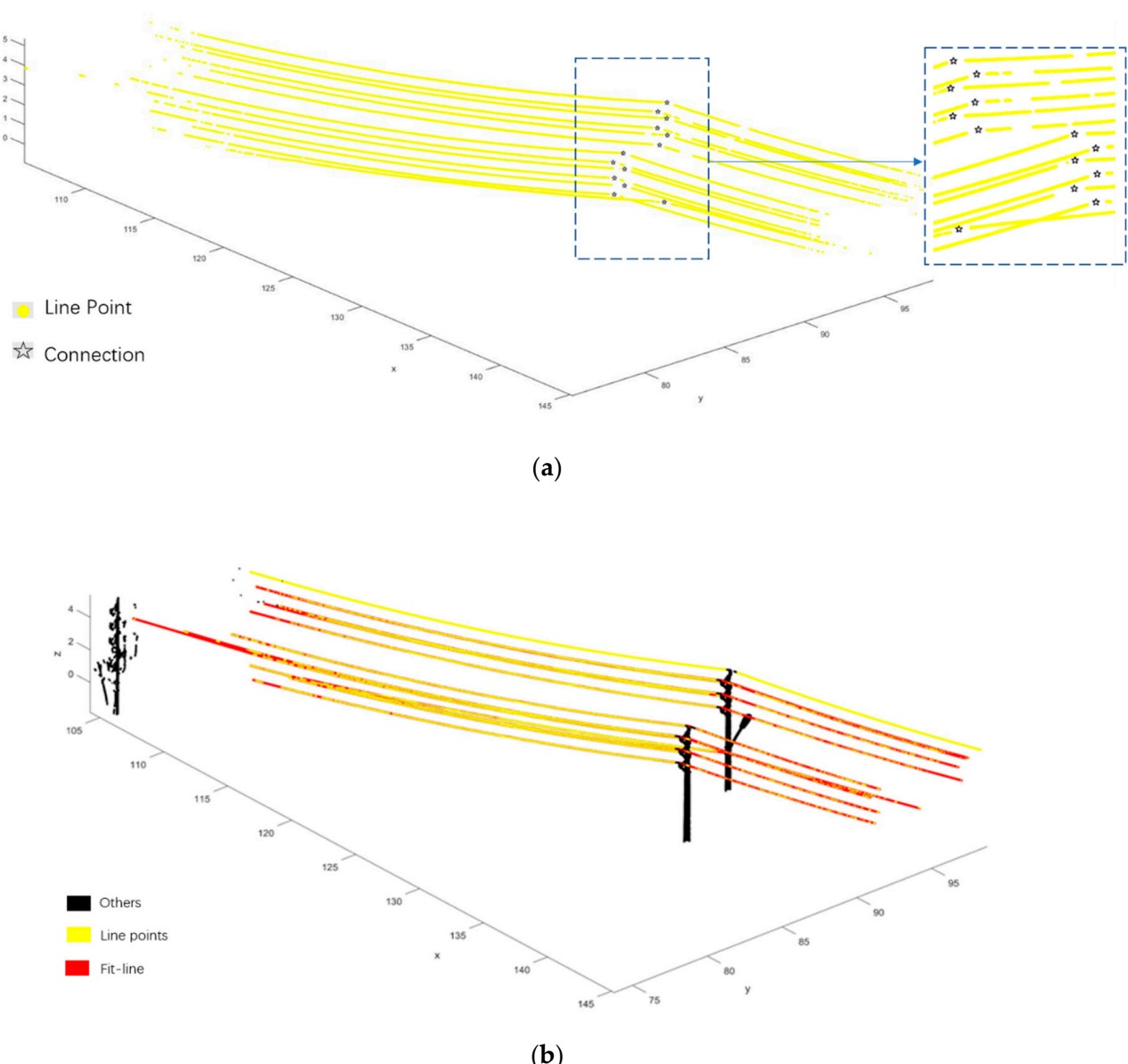

(a)

(b)

**Figure 15.** Connection finding and powerline reconstruction. (**a**) Pylon–powerline connections finding. Connections are shown as black stars and powerline points are colored in yellow. (**b**) Reconstruction results. Outliers are colored in black and powerline points are in colored in yellow. Fitted results are presented by red lines, most of which are obscured by line points.

### 4.3.2. Parameters of Powerline Candidate Selection

Due to the characteristics of low–voltage transmission lines and the change of point cloud density, $r_d = 0.3$ can separate the power lines separately [11]. Therefore, we set $r_d = 0.3$ to obtain the powerline clusters correctly. According to the State Grid's regulations on the construction of 35 kv transmission lines, the distance from the outer edge of the insulation layer to the center of the conductor cross–section is approximately 0.01 m. Appropriate values can distinguish the powerline candidate from the others. Considering that the powerline candidate is not a strict straight line in a natural environment, which also makes its cross–section not a standard circle and results in a larger $sp_v$, we set the $r = 0.02$ m and $sp_t = 0.8$ as empirical values after several experiments to ensure that there is no misjudgment in the process of selecting power line candidate points.

### 4.4. Quantitative Evaluation

In this section, effect of powerline extraction and reconstruction are evaluated from the perspective of data. Precision, recall and $F_1$-*score* are used to evaluate the results of the hierarchical clustering method, and they are computed by following formulas:

$$Precision = \frac{T_P}{T_P + F_P} \tag{7}$$

$$Recall = \frac{T_P}{T_P + F_N} \tag{8}$$

$$F_1 - Score = \frac{2 \times Precision \times Recall}{Precision + Recall} \tag{9}$$

where $T_P$ denotes the number of true positive points, which is the number of powerline points found in both ground truth and detected data. $F_P$ is the number of false positive points, which is the number of powerline points that were detected but did not exist in ground truth. $F_N$ denotes the number of false negative points, which is the number of powerline points found in ground truth but were not found in detected data.

Table 3 lists the average percentage of recognition precision, recall, and $F_1$-*score* in the hierarchical clustering method under different gap widths, where gap width 0 represents the test data without manual gaps. The results show that the proposed method has a high precision, recall, and $F_1$-*score* in individual powerline extraction, which indicates that the method can extract most powerline points correctly. In addition, results have little change even when the gap width is increased to $12s_l$, which proves that the hierarchical clustering method has high robustness in processing gaps.

**Table 3.** Evaluation of powerline extraction.

| Gap Width (m) | Average Precision (%) | Average Recall (%) | Average $F_1$-*Score* (%) |
|:---:|:---:|:---:|:---:|
| 0 | 100 | 98.3 | 99.1 |
| $s_l$ | 100 | 98.3 | 99.1 |
| $2s_l$ | 100 | 98 | 98.9 |
| $4s_l$ | 100 | 97.5 | 98.5 |
| $8s_l$ | 100 | 97.3 | 98.2 |
| $12s_l$ | 100 | 97.1 | 98.1 |

For powerline reconstruction, the coordinate difference (Error) between estimated connections and the connections in ground truth (real connections found manually in obtained TLS data) is used to evaluated effect of the slope change method, and RMSE is used to evaluate the fitting results. Table 4 lists the analysis of the data results. The highest fitting accuracy is 0.012 cm, the lowest is 0.021 cm, and the average fitting accuracy is 0.015 cm. In summary, the overall accuracy of powerline fitting is high, which proves the feasibility of the pylon–powerline connection finding method.

**Table 4.** Evaluation of connection finding and reconstruction.

| Line | X_Error (cm) | Y_Error (cm) | Z_Error (cm) | RMSE (cm) |
|:---:|:---:|:---:|:---:|:---:|
| 1 | 2.89 | 3.80 | 2.71 | 0.019 |
| 2 | 2.76 | 3.51 | 2.23 | 0.018 |
| 3 | 2.74 | 3.09 | 1.99 | 0.017 |
| 4 | −0.87 | −1.12 | −0.61 | 0.012 |
| 5 | 0.76 | 1.22 | 0.34 | 0.012 |
| 6 | 0.51 | 1.04 | 0.29 | 0.012 |
| 7 | 2.87 | 2.95 | 2.13 | 0.017 |
| 8 | 1.33 | 2.02 | 0.98 | 0.013 |
| 9 | 1.71 | 2.37 | 1.38 | 0.013 |
| 10 | 2.91 | 4.17 | 2.60 | 0.019 |
| 11 | 3.84 | 4.21 | 3.41 | 0.021 |
| 12 | −1.56 | −1.83 | −0.74 | 0.013 |
| 13 | 1.28 | 2.43 | 0.83 | 0.013 |

## 5. Conclusions

Powerline detection is important for powerline monitoring, previous studies had several attempts at powerline extraction based on LiDAR point cloud data. However, due to the increasing distance from the scanner or occlusion caused by the presence of other objects, gaps will appear in LiDAR point clouds, which makes it difficult for methods that rely on continuous relations (such as region growth or EC) to process powerlines with gaps. To solve this concern, a robust method is proposed to repair gaps and extract powerlines from TLS data, which can create continuous neighborhood relations by estimating the centroids of gaps and extract individual powerlines directly without any prior conditions. For verifying the robustness of the proposed method, we implemented the method using data with gaps of different widths. Experimental results showed that the method can extract individual powerlines even when the gap width increased to $12s_l$ (6 m), and the recognition precision changed little as the gap width changes. What is more, the ability to handle large gaps ensures that the proposed method can be applied to various scenarios with gaps, which provides great potential for repairing gaps in the line extraction field. Although the robustness of the proposed method has been proved, the recognition precision needs to be further verified under complicated environments in our future work.

In addition, we reconstructed the powerlines of two spans using pylon–powerline connections found by the proposed slope change method. Results showed that the found connections are of small error, which validates the feasibility of the method in finding connections. In summary, this method has certain application value for multi–span powerline reconstruction in the future work.

**Author Contributions:** Conceptualization, Y.F. and R.Z.; methodology, Y.F. and R.Z.; software, Y.F. and X.F.; validation, X.F. and R.D.; formal analysis, Y.F. and R.Z.; investigation, X.F. and R.Z.; data curation, M.X., X.F. and R.D.; writing—original draft preparation, Y.F. and R.Z.; writing—review and editing, Y.F. and R.Z.; visualization, Y.F.; supervision, R.Z. All authors have read and agreed to the published version of the manuscript.

**Funding:** This research was financially supported by the Ministry of Education Joint Fund (No. 6141A02011907), the National Natural Science Foundation of China (NSFC) (Grant Nos. 41674017), and the National Key Research and Development Program (Nos. 2016YFB0501803 and 2016YFB0502202).

**Institutional Review Board Statement:** Not applicable for studies not involving humans or animals.

**Informed Consent Statement:** Not applicable for studies not involving humans or animals.

**Data Availability Statement:** The data presented in this study are available on request from the corresponding author.

**Acknowledgments:** We would like to thank Jingren Wen (GNSS Research Center, Wuhan University) for providing data, and we would also like to thank four anonymous reviewers for technical support.

**Conflicts of Interest:** The authors declare no conflict of interest.

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
