# Peer review of "A Hierarchical Clustering Method to Repair Gaps in Point Clouds of Powerline Corridor for Powerline Extraction"

_remotesensing, doi:10.3390/rs13081502_

Round 1

Reviewer 1 Report

The issue of automatic extraction of power lines is very relevant and topical, so the paper is interesting. Unfortunately, many links and references are missing so int's not possible to fully evaluate the paper in the current form.

Although I don't think I am fully qualified to judge the use of the English language, there are unintelligible expressions. Among others, "tremendous(?) points" (p. 2), or "beautiful yet incomplete (?) , few powerline points..." (p.17). For this reason, a complete revision of the language is suggested.

In the introduction, the term 'camera sensor' is used, when it would be more appropriate to speak of a photogrammetric method. Incidentally, the SfM/Mvs photogrammetric method also currently acquires spatial information in the form of a dense point cloud and are used for powerlines detection and segmentation. For example, the latest versions of Agisoft Metashape have a "detect powerlines tool".

As power lines detected by the photogrammetric method also suffer from gaps and defects, so the method described could be tested and applied to them.

Reviewer 2 Report

The paper ‘A Hierarchical Clustering Method to Repair Gaps in Point Clouds of Powerline Corridor for Powerline Extraction’ is generally written in correct English with minor editorial errors. The study is interesting and includes in the journal scope, but the experiment design description needs improvement. The mathematical model is only partially presented, eg. in the paper there is only one authors’ equation related to data processing (#1 in the paper), and three QC formulas (#5-7). As a result, the methodology is not clear and the summary of the work is not convincing. Additionally, the discussion section did not refer to other studies in this area, although other researchers focused on similar topics. There are several issues in the study that require better data-mining, as the authors sometimes introduce unjustified simplifications. Details in the remarks below.

I suggest conditional acceptance of the paper after major revision.

Remarks:

  • Citation errors in introduction (e.g. p. 1, 4, 5, 12)
  • Rotation: „we manually calculated the average angle between powerlines and X-axis from XOY” – what is the accuracy of the manual approach? Least-squares adjustment of planar (2D) powerlines coordinates should be considered instead, to obtain the direction estimator accuracy.
  • Powerlines clustering method using sections (section 3.2.1): " main direction of powerline cluster tends to be horizontal." - what if there are significant differences in the height of successive pylons or a significant distance between the pylons? There is a significant curvature of powerlines in that cases. It is not justified to accept the horizontal direction of the cross-sections. Instead, it is necessary to (1) determine the slope using the LS method in the plane corresponding to the direction of the powerline, (2) rotate the point cloud segment around the axis perpendicular to the plane of the powerline run, (3) then adopt the method of horizontal cross-section analysis.
  • Powerline curvature determination (section 3.3): „Therefore, it is unrealistic to fit powerline with one model” – the sentence raises serious doubts. For example, the cited work [13] presents the method fitting of a polynomial into the powerline section. Hence, accepting the generalization of the linear course of the end parts of the powerline with the pylons is not acceptable. It is necessary to determine curvilinear powerline section parameters and on their basis calculate the point of intersection. Please note, that the authors used the related method for missing cloud points generation using the quadratic polynomial.
  • Inadequately justified selection of parameters, for example: “In this paper step length.is set to 0.5m”. Why this value and not, for example, 0.4 or 0.6 m. The results of the analysis of competing variants of the radius length should be presented and based on it, the final value should be selected. Second example: " Lots of experiments have shown that setting slope threshold to 0.3 can successfully separate powerlines from other objects." - the studies where this value has been adopted should be indicated.
  • Metrological error in Table 3 in the RMS column. What is the reliability of RMS given with 1e-8 or 1e-9 cm precision? Since all the values in the table are 1e-2 cm precision, then a reliable RMS should have a precision of 1e-3 cm. In the described case, since RMS obtained the values of 1e-4 and 1e-5 cm, it should stop at two significant numbers.

Reviewer 3 Report

The paper presents a method for segmentation and reconstruction of powerlines from TLS data. Although the work is original and shows good results, there are many things to improve. Most importantly:

  • Some explanations are confusing and the language is not very technical.
  • Figures could be improved
  • Some sections seem a little bit disorientated, intentionally

Other issues:

"The difference is that MLS scans close to targets, so it can acquire data of higher precision." Complete "The difference with ALS is..." But here the main thing about mls, and it should be mentioned, is that mls is subject to the fact that it needs to travel on roads, which sometimes do not exist underneath the powerlines.

"Shen et al.[12] put point cloud". Shen et al.[12] structure point cloud

The term cross section is clear and can be used more often

"Experiment results show that the method proposed in this paper can successfully extract powerline in the presence of gaps with high robustness." This is a conclusion, here the authors should indicate expected benefits, not obtained from the results.

"Data Description" Case study better.

It is advisable to include a map that positions the case study, or road references, mileage, GPS coordiantes.

The authors talk about several scans, but in the results, the cloud only shows one scanning position. Can you clarify this?

If authors used several scan positions: exact number of scans, number of points per scan, maybe an illustration of the scans would also be useful.

"Some references are missing on text" Error in references

"The LiDAR was fixed on a tripod for stationary scanning. Compared to ALS, TLS has higher precision but more occlusion, which is appropriate for experiment in this article." Delete

In figure 1a nothing is visible, zoom in.  In general, many figures do not look good and take up a lot of white space.

"What's more" this type of expression is very emphatic for a technical document.

"TLS data can be used as provisionally appropriate test data to verify the gap repair method because of its serious occlusion problem" It is not clear

"The input powerline point cloud data is directly clustered into individual lines" When?

Deletea arrows from second Figure 1. With colourbar is enough

Figure 2a, 2c, 2d can be deleted. 

Figures 2e and 2f can go in a separate figure, indicat XYZ axes y both.

It is also not recommended to explain the method in each figure caption, where only the figure should be explained.

In general, variables should be in lower case and italics, upper case should be reserved for arrays and matrices. 

Figure 4a can be deleted

Test data mus be described in results, or case study. It is better do not mention it methodology. Test data it is no clear.

"The clustering method is carried out segment by segment and begins with a starting segment." Improve

If the starting point is where most of it is, and it only goes in one direction, can it happen that it starts in the middle and is scanned without matching?

Figure 5. The zoom in is not enough. Colors are not visible

The first paragraphs of Section 4.1 are typical of a comparison of results that should be made after showing results.

"Beautiful yet incomplete" does not sound profesional

Table 2 is numered as table 3

It is not clear how is generated the ground truth for reconstruction

"Further, due to the similarity of gaps, hierarchical clustering method can also be applied in ALS or MLS data" This is very risky, it depends on the density, if you want to affirm it you should test by reducing the density of the data.

"It is believed that" Personal opinion, delete.

Finally, regarding gap filling, although it is the most innovative aspect of the paper, its relevance is not clear. If the gap is smaller than Si, does it regenerate? Do gaps smaller than Si affect the rest of the process? Moreover, there is no quantification of the outcome of regeneration. In that respect, I suggest to the authors to generate small artificial gaps, that is to regenerate them and estimate the precision with the original cloud.

Round 2

Reviewer 2 Report

All necessary corrections were introduced.

Reviewer 3 Report

The authors have done a great job in improving the article. All the answers have been thorough and rigorous. I recommend the acceptance of the article

Just two last comments:

  • In figure 1 it would be necessary to locate in the figure on the right the exact position of the scan. By the way, in the manuscript figure 1 is cut off.
  • In future revisions, a good practice is to mark in the manuscript the changes made.